# Development, Simulation of Temperatures, and Experimentation in Injection Molds Obtained through Additive Manufacturing with Photocurable Polymeric Resins

**DOI:** 10.3390/polym15051071

**Published:** 2023-02-21

**Authors:** Adrian Benitez-Lozano, Carlos Vargas-Isaza, Wilfredo Montealegre-Rubio

**Affiliations:** 1Grupo de investigación Calidad Metrología y Producción, Instituto Tecnológico Metropolitano, Medellín 050034, Colombia; 2Grupo de investigación Diseño y Optimización Aplicada, Facultad de Minas, Universidad Nacional de Colombia, Medellín 050034, Colombia

**Keywords:** rapid tooling, injection molding, stereolithography (SLA), photocured resin

## Abstract

Additive manufacturing (AM) is a relatively new option in mold manufacturing for rapid tooling (RT) in injection processes. This paper presents the results of experiments with mold inserts and specimens obtained by stereolithography (SLA), which is a kind of AM. A mold insert obtained by AM and a mold produced by traditional subtractive manufacturing were compared to evaluate the performance of the injected parts. In particular, mechanical tests (in accordance with ASTM D638) and temperature distribution performance tests were carried out. The tensile test results of specimens obtained in a 3D printed mold insert were better (almost 15%) than those produced in the duralumin mold. The simulated temperature distribution closely matched its experimental counterpart—the difference in average temperatures was merely 5.36 °C. These findings support the use of AM in injection molding and RT as an excellent alternative for small and medium-sized production runs in the global injection industry.

## 1. Introduction

The use of core-cavity stereolithographic assemblies in low-volume injection molding is experiencing steady growth worldwide. Nevertheless, employing plastic instead of metal molds poses several problems in terms of mold handling, material injection, and process requirements [1]. In 1986, stereolithography (SLA) was the first technology in the market of 3D printing or additive manufacturing. In SLA, a liquid resin is transformed into a solid material by exposing it to ultraviolet (UV) light. The raw material is a photopolymer, which is cured by UV light using a build platform that moves up and down in order to create small spaces of liquid resin between the surface of the printer and the part that is being made. As shown in Figure 1, the printing platform is brought as close to the bottom or surface of the resin as possible, and UV light is used to photo-cure, burn, or sinter the shape of the part in its final form.

A variation of SLA is sintering or curing the resin using a liquid crystal display (LCD) to print the part. In LCD printing, an LCD screen projects an image onto the lower surface of the bed, thus sintering and curing the piece.

This paper is structured as follows. Section 1 presents the features and characteristics of the operation of SLA. Section 2 reviews the state of the art of this AM technique applied to rapid tooling (i.e., part design and manufacturing). Section 3 describes the methods, equipment, and materials used here to make mold inserts for the production of ASTM D638-compliant specimens. Section 4 reports and analyzes the results of stress tests and (single-point) thermal simulations versus experimentation. Finally, Section 5 draws the main conclusions of this experimentation.

Another study investigated the production of injection molds (prototypes) by three different technologies: stereolithography (SLA), laser sintering (LS), and 3D resin photopolymerization (e.g., PolyJet). To validate the use of these three technologies in injection molding, different materials were evaluated to find the most suitable option for each [2].

Other authors implemented SLA with photocurable resins to achieve fast injection tooling and evaluate the mechanical and thermal properties of commercial polypropylene. As their results indicate that the nature of the resin inserts affects the crystallinity of the mold parts in terms of microstructure, their recommendation is to use the molds in short production runs and pilot tests [3]. The influence of thermomechanical loads on the lifecycle of mold inserts produced by additive manufacturing can be a differential factor that should be studied to improve the performance of these tooling technologies [4].

Different studies have characterized material microstructures in order to create simulation models based on information about material properties obtained by multiple techniques, e.g., differential scanning calorimetry (DSC), thermomechanical analysis (TMA), and scanning electron microscopy (SEM). To elaborate on the ideas presented in this introduction and delve deeper into the literature on this topic, the next subsection describes the state of the art of SLA in the industry of RT mold injection.

## 2. State of the Art of SLA in Rapid Tooling

Several authors agree: injection molds and tooling are expensive, and obtaining them is only economically viable when high levels of production are assumed [5,6,7,8,9,10]. They have studied additive manufacturing techniques for making molds and tooling, i.e., by stereolithography (SLA), laser sintering (LS), and 3D resins photopolymerization (e.g., PolyJet), and evaluated different process parameters during injection cycles [2].

Some of them evaluated the performance of molds made with reticulated structures based on titanium, aluminum, and vanadium alloys. In their case, they used computer-aided engineering (CAE) simulations to observe the behavior of these lightweight structures (up to 80% lighter than traditional molds). In their experiments, they achieved 400 complete production cycles in an injected polyvinyl chloride (PVC) material [11].

Others investigated the microtexture of molds obtained by SLA that are employed to manufacture medical components, evaluating the effect of SLA on mold surfaces. They studied the tribological properties of said microtexture and their impact on some biomedical implants and found that the printing direction can improve the surface properties of the piece [12].

Zirconia has also been used as a stabilizing material for molds obtained by SLA [6] and its derivative, i.e., digital light processing (DLP), which is the SLA approach adopted in this study. Nowadays, several industrial processes are already using additive manufacturing to obtain molds for short production runs and cycles as an alternative to tooling by traditional manufacturing [3,13,14].

In recent years, many companies have migrated towards additive manufacturing using injection molds to produce components. In fact, some design and research teams have managed to produce up to 80 components using PolyJet and SLA technologies. The next challenge for these teams is to introduce their products into the market [15]. Rapid tooling, such as additively manufactured mold inserts, has great potential in the context of mass customization as it combines the strengths of traditional mold manufacturing (in terms of production) with the flexibility offered by additive manufacturing in mold inserts [16].

A study implemented additive manufacturing of inserts by extrusion in order to maximize the number of cycles before failure (due to high injection pressures and clamping force). The mechanical properties of the inserts were improved, achieving 15 successful cycles before the presence of insert cavity deformation caused by accumulated injection pressures [17].

Very few papers have analyzed mold inserts obtained by SLA. Nevertheless, one of them did report results obtained by tooling using photocured polymers (which was performed in a way that is similar to the method in this study). That study found that the nature of the inserts affected the crystallinity of the parts, but, in terms of mechanical properties, said inserts were similar to parts molded with steel tooling. This indicates that these systems and inserts obtained by additive manufacturing can be used in industrial manufacturing. Pilot tests have been conducted in applications where these properties are critical for short production runs [6].

As described in Section 1 and Section 2, SLA is very important for the injection industry because it can produce easy, medium, and complex geometrical designs and cavities for injection molding. It can also be used to develop and manufacture interchangeable inserts, which can reduce the time it takes to change formats according to the SMED (Single-Minute Exchange of Die) philosophy. Based on the results reported in Section 4 of this paper, the mechanical properties of the parts were improved, which is very important for the final quality and product lifecycle. All these aspects had not been considered together in any of the articles reviewed in this section, which constitutes the research gap addressed in this study.

## 3. Materials and Methods

A Photocentric LC Magna, from manufacturer Photocentric LTD in Peterborough (UK), stereolithographic 3D printer was used for additive manufacturing. This equipment can 3D print up to 15 kg of resin and make customized geometries for all industries. In this case, the printed inserts and cavities were adapted to rapid tooling for plastic injection molding. Figure 2 shows the workflow of additive manufacturing in this printer.

The mold insert obtained here was made of Photocentric HighTemp DL400 resin (Photocentric LTD in Peterborough, UK), a photocurable resin with excellent thermal and mechanical performance. It has remarkable properties in terms of resistance to impact, compression, fatigue, high temperatures, and humidity, as well as mechanical rigidity without presenting deformations.

Different layer thicknesses (50, 100, 200, and 300 µm) were tested, and they produced the same result in surface quality and good definition. However, low-layer thicknesses require longer manufacturing times. Therefore, a layer thickness of 350 µm was selected to achieve printed parts in shorter times, which makes this process a suitable alternative for fast tooling and developing injection mold inserts. The 3D printing parameters in Table 1 were selected to achieve the best quality in the mold insert (i.e., good tolerance on the surface of the piece, and good filling) in the shortest possible time.

Table 2 details the main mechanical properties of the HighTemp DL400 resin used in this study.

In addition, polypropylene specimens were manufactured by injection molding in molds obtained by additive manufacturing and traditional molds made of duralumin. The properties of such polypropylene are shown in Table 3.

The tensile tests were conducted on a Shimadzu/AG 100 kNX universal testing machine, from manufacturer Shimadzu Corporation in Kioto, Japan, with a 10-kN load cell. All the specimens obtained in this study were injected using a WELLTEC TTI-90F2V horizontal injection molding machine, from Welltec Machinery Limited in Hong Kong, China, with a 90-ton clamping force. Table 4 shows the injection parameters employed here for this equipment, which are presented by feeding, compression, and dosing zones; pressures and velocities in each of them; and holding parameters.

These injection parameters were tuned to fill the volume of the cavity completely. This was achieved by programming the volume of molten plastic (in the plasticizing cylinder of the injection molding machine) based on the length of the cylinder, the pressure, and the velocity of the screw of the injection molding machine (see Table 4). By adjusting these parameters according to the results of preliminary injection tests, it is possible to obtain a complete filling of the mold cavity with homogeneous temperatures of the molten plastic. The holding parameters make it possible to inject a piece without noticeable contractions and to achieve a constant weight. In this study, these parameters were adjusted using computer simulations and preliminary experimental evaluations of injected parts’ weights. The necessary clamping force to be applied by the machine (determined based on pressures) did not exceed 90 tons, which is its clamping capacity.

Figure 3 details the geometry of the injected specimens to be obtained using mold inserts produced by SLA (i.e., additive manufacturing). Said inserts are similar to those obtained by rapid tooling or molds made by conventional machining (made of duralumin). This geometry complies with ASTM D638 (Standard Test Method for Tensile Properties of Plastics) [20]. The cavity mold was designed to produce two parts for tensile testing in a single injection cycle. The feeding system parts do not interfere with the quality of the piece because they are located at the ends of the specimen (top and bottom on the tensile tester), that is, the areas of contact between the specimen grips of the tensile testing machine and the part.

Finally, an EXTECH SD200 3-Channel Temperature Datalogger (i.e., a contact thermocouple) was located at the bottom of the right cavity to measure the experimental temperatures at one point in the mold cavity. This configuration enabled us to measure temperatures at three points in a period of time of up to 600 s, with 5 s time steps. Table 5 presents the general specifications of this equipment.

The filling and cooling stages of the mold were simulated in Altair^®^ Inspire™ Mold software with the following parameters: pressure, 6 MPa; pressure holding time, 5 s; maximum compaction, 10 s; total cycle time, 72; filling time, 60 s; and flow rate of 17,016.4 mm^3^/s. A mesh of tetrahedral elements with a size of 4.6 mm for each element and an aspect ratio of 99.43% was generated, for a total of 2096 elements and 962 nodes, which is arguably a good mesh quality for all the measurements. The boundary conditions were implemented in the cavity zone of the inserts under two thermomechanical and physical conditions: a clamping force of 0.8 tons and a temperature of 493 K.

## 4. Results

The tensile tests were carried out using ASTM D638-compliant specimens of three kinds: (i) DL400 resin specimens obtained by SLA, i.e., additive manufacturing; (ii) polypropylene specimens obtained in a 3D printed DL400 polymer resin mold inserts; and (iii) polypropylene specimens obtained by injection molding inside a traditional duralumin mold produced by subtractive manufacturing.

### 4.1. DL400 Resin Specimens Obtained by SLA

The tensile tests were carried out in accordance with ASTM D638-14 (Standard Test Method for Tensile Properties of Plastics). Figure 4 shows pictures of the DL400 resin specimens obtained by SLA (additive manufacturing) before and after failure.

According to ASTM D638-14, five tensile tests were carried out. Nevertheless, an additional specimen (T1) was tested for parameterizing and fine-tuning the variables. Test specimen 3 (T3), in Table 6, presented a very low tensile stress value (an outlier), probably due to the brittle behavior of the resin in the grip area. The standard for this test establishes that the specimens should be prepared in advance by eliminating marks on their surface using an abrasive material. Table 6 summarizes the results of the tensile tests on the specimens produced by SLA 3D printing.

Table 6 describes the mechanical performance of the DL400 resin used in the experiments to establish a point of reference for its mechanical properties. The data sheet of this resin indicates a maximum tensile strength of 80 MPa if it has not been processed [18]. According to the results in Table 6, the maximum tensile strength of the resin after 3D printing was only 26.6 MPa. Therefore, it is very important to compare this property in the three kinds of specimens studied here: (i) DL400 resin specimens obtained by SLA; (ii) polypropylene specimens obtained in 3D printed DL400 polymer resin mold inserts; and (iii) polypropylene specimens obtained by injection molding inside a traditional duralumin mold.

### 4.2. Polypropylene Specimens Obtained in a 3D Printed DL400 Polymer Resin Mold Insert

The Photocentric LC Magna stereolithographic 3D printer described in Section 3 was used to make the injection mold inserts shown in Figure 5a and Figure 6 (for initial experiments). Figure 5a shows the runner and feeding system. Figure 5b is a picture of the assembled injection molding system showing some flash, which should be improved with more experimentation. Figure 5c,d are pictures of the parts injected employing said system. Table 3 in Section 3 (Methods and Materials) details the parameters of this injection process. The initial results of the SLA printed resin specimens and mold inserts show that SLA can produce parts and tooling that fulfill functions in a traditional manufacturing process (e.g., plastic injection molding).

To establish the performance of the mold inserts in terms of lifecycle and the number of pieces produced in initial experiments, another similar mold insert made of resin (as shown in Figure 6) was tested. After 85 full injection cycles, the insert presented a crack in the upper part, as can be seen in Figure 6. This was probably a consequence of the injection pressure and clamping force, which could have generated cracks in the areas surrounding the feeding channels of the mold. Additionally, the mold insert was fragile because its impact resistance was 15.6 J/m according to Test Method C [19], as detailed in Table 2. Test Method C is applied to very brittle plastics whose Izod impact resistance is less than 27 J/m.

The improved mold insert in Figure 5a was tested in subsequent experiments but did not exhibit damages or cracks on its cavity surface. This version of the mold was functional, and its lifecycle was better than that of the first version (Figure 6) because it completed more than 85 full cycles in an experimental production run. Table 7 presents the consolidated results of the tensile tests (carried out according to the standard mentioned above) on the polypropylene specimens obtained by injection into a resin mold insert produced by SLA (i.e., additive manufacturing).

The values in the table above are represented in Figure 7 using a bar chart with error bars to examine the variability of the data obtained in the measurements of each one of the stress tests carried out on the injected specimens. These results will be compared to those of the specimens obtained by additive manufacturing only and those of the specimens obtained by injection into duralumin molds (as presented in Section 4.1 and Section 4.3, respectively).

The error bars in Figure 7 (for one each of the tests) demonstrate that the data did not exhibit great variability, which supports the validity of the test, as well as the reliability of the measurements and the results obtained in this study. 

### 4.3. Polypropylene Specimens Obtained in a Duralumin Mold

Five polypropylene ASTM D638-compliant specimens were obtained in a duralumin injection mold produced by conventional machining. The experimental results indicate that these specimens had lower tensile strength and better surface texture than those obtained in the photocurable resin injection mold insert. Table 8 shows the results of the tensile tests carried out on these five samples.

Figure 8 presents these strain results in a bar chart with error bars because it is important to verify the variability of the data.

The tensile strain of the specimens obtained in a duralumin mold was reduced by 15.5% compared to that of their counterparts obtained in a mold that contained a resin insert, as shown in Table 7 and Figure 8. As previously mentioned, this can result in better mechanical performance, i.e., tensile properties, of specimens obtained in mold inserts produced by SLA (additive manufacturing). The simulations conducted in this study could have predicted this improvement in terms of thermal properties and temperature distribution, as shown and explained in the next subsection. In particular, said improvement can be predicted and simulated as a model with linear properties. This is a significant conclusion of this study regarding the material model of the resin used here. Furthermore, the values of mechanical properties that were found are highly reliable because the requirements and specifications established in the ASTM D 638 standard were rigorously applied.

### 4.4. Simulation and Experiments of Single-Point Temperature Profiles

To analyze the behavior of the temperatures at a point on the mold cavity in simulations and the respective experiments, it was necessary to adapt the instrumentation of the physical mold with the thermocouple described in Section 3 and the characteristics listed in Table 5. The simulations and conditions presented in Section 3 were applied to present the results in the cooling stage of the mold. In this case, Figure 9 shows the temperatures in two areas of the same specimen: the specimen surface (top) and the area in contact with the mold cavity and the inserted thermocouple (bottom).

The simulations reported an average temperature of 355.43 K (or 82.28 °C), as specified in Table 9. These data were calculated by placing a virtual thermocouple right at the spot of the measurement on the cavity, as shown in Figure 9. Table 9 also includes a third column with the equivalent simulated temperature in degrees Celsius. These temperatures were simulated and measured at the end of the cooling stage of the injection cycle.

Figure 10 shows the CAD design of the mold, the assembly with the contact thermocouple, its location at the bottom of the mold, and the specimen measurements (in mm).

The instrumentation presented in Figure 10b was used to collect data on the behavior of the temperature profile at one point in an area of the cavity, as indicated by the arrows in Figure 10b. The data obtained by the thermocouple were recorded every 5 s, for a total of 220 measurements over 24 production cycles, as shown in Figure 11.

The highest values (temperature peaks) in Figure 11 were listed in Table 10. They represent active zones of molten and later solidified material in the cavity. These values can be compared to those that were simulated and reported above in Figure 9 and Table 9. In particular, Table 10 details the highest experimental temperature measurements tagged with an ID number. The average temperature in this experiment was 76.92 °C.

The difference in average temperature between the simulations and the experiments with actual specimens (Table 9 and Table 10, respectively) was only 5.36 °C, which represents a small margin between predictions and experiments.

## 5. Conclusions

One of the many advantages of producing plastic injection inserts and tooling by stereolithography and additive manufacturing is the feasibility of obtaining complex geometries and cavities that could not be developed using conventional manufacturing. In addition, the production times of mold inserts produced by SLA are significantly shorter.

In this study, the photocurable resin presented an acceptable behavior in terms of mechanical properties (e.g., melting point, tensile strength, and hardness), as demonstrated in Figure 5 and Figure 6.

The specimens injected into an SLA-printed resin mold insert (produced by additive manufacturing) presented a slightly higher mechanical resistance than those obtained by SLA directly and those injected into a duralumin mold. The increases were 22.5% and 15.5%, respectively. This could be due to changes in the crystalline microstructure of the material when injected. These results are acceptable in terms of the functionality of the part and the basic characteristics that it should have.

Based on the total number of full production runs that are necessary to inject the specimens, the feeding system of the polymeric resin mold should be redesigned so that the material flows better and the pressures and clamping forces that are generated can be alleviated. This can reduce cracks and microcracks that trigger material fracture.

## Figures and Tables

**Figure 1 polymers-15-01071-f001:**
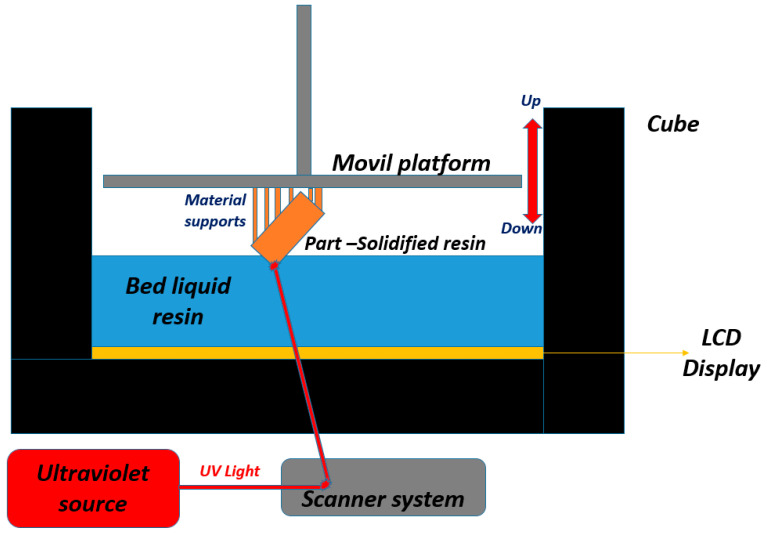
Printing principle of stereolithography (SLA).

**Figure 2 polymers-15-01071-f002:**
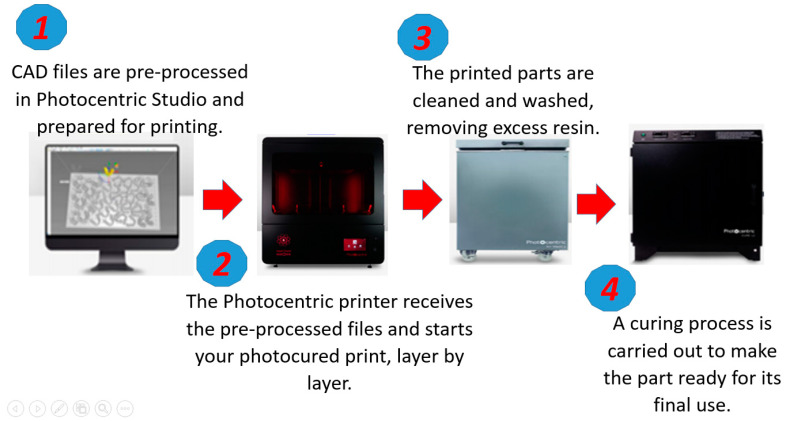
Printing workflow of the LC Magna stereolithographic 3D printer.

**Figure 3 polymers-15-01071-f003:**
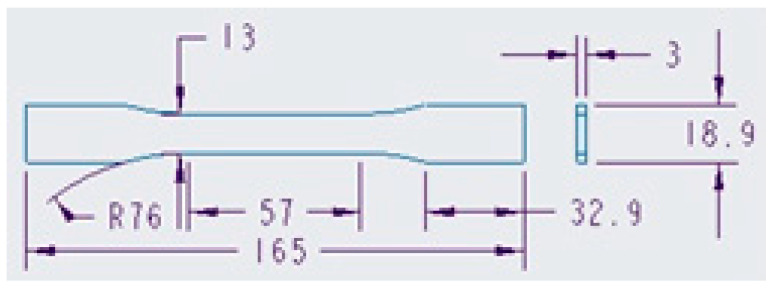
CAD design (in mm) of all the specimens that were subjected to tensile tests. This was the design of the three groups of specimens tested in this study: (i) DL400 resin specimens obtained by SLA; (ii) polypropylene specimens obtained in 3D printed DL400 polymer resin mold inserts; and (iii) polypropylene specimens obtained by injection molding inside a traditional duralumin mold produced by subtractive manufacturing.

**Figure 4 polymers-15-01071-f004:**
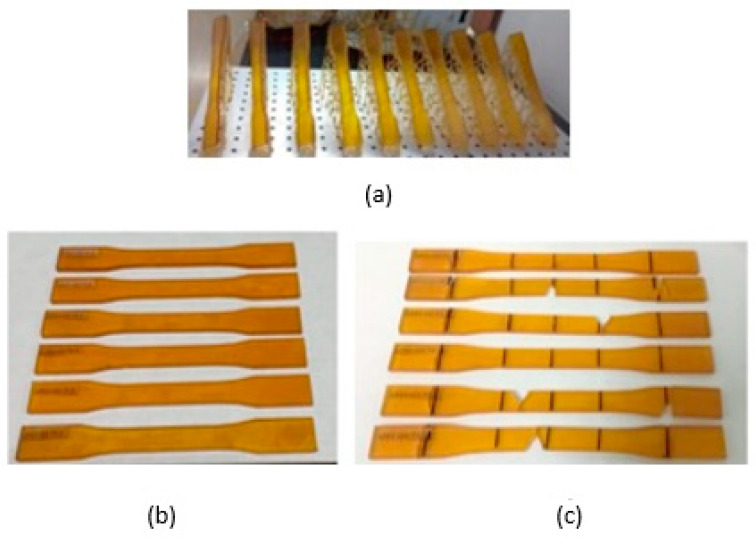
(**a**) ASTM D638 specimens made of DL400 resin obtained in a 3D printer. (**b**) Specimens before failure. (**c**) Failed specimens showing brittle fracture.

**Figure 5 polymers-15-01071-f005:**
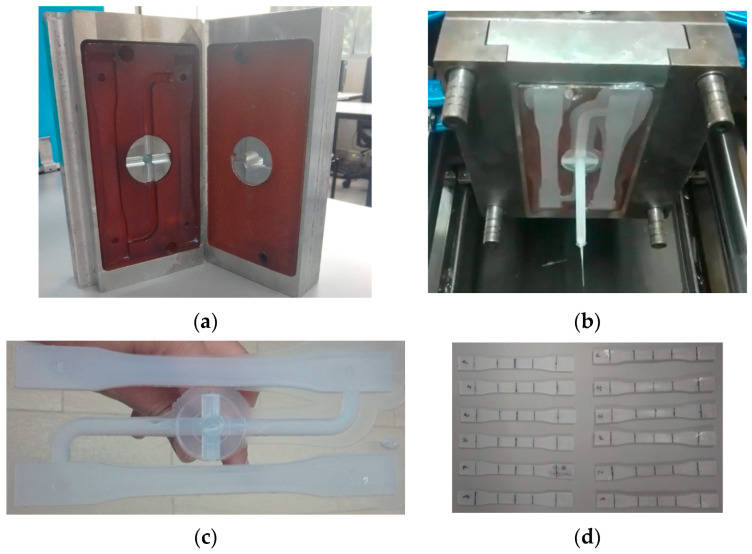
(**a**) Resin mold inserts obtained by additive manufacturing in a Photocentric LC Magna 3D printer. (**b**) Polypropylene specimens injected into the mold with inserts produced by additive manufacturing. (**c**) Polypropylene specimens and feeding system obtained (100% filling). (**d**) Functional polypropylene specimens for ASTM D638 tensile testing.

**Figure 6 polymers-15-01071-f006:**
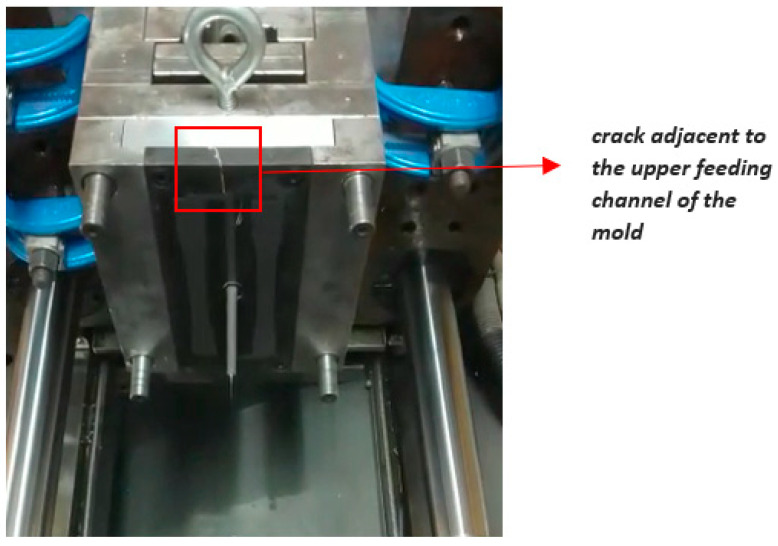
A crack appeared in the top area of the mold insert, near the upper feeding channel.

**Figure 7 polymers-15-01071-f007:**
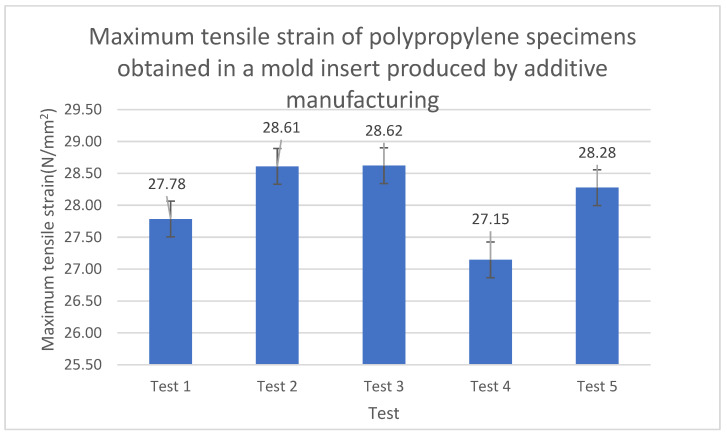
Maximum tensile stress of specimens obtained in the SLA printed mold insert.

**Figure 8 polymers-15-01071-f008:**
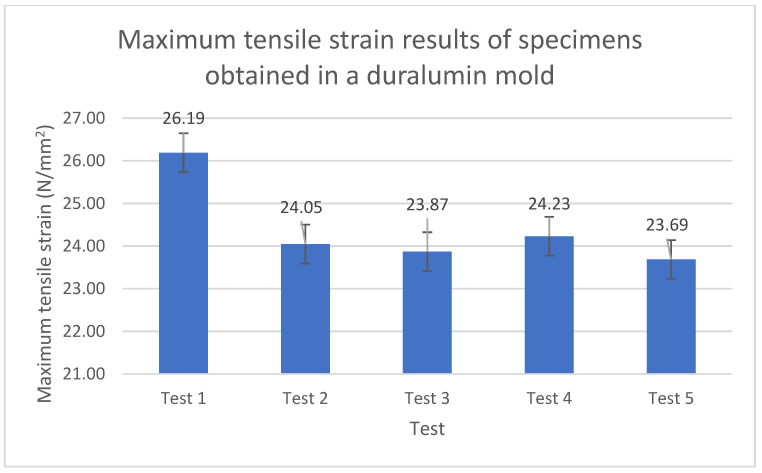
Maximum tensile strain of the specimens obtained in a duralumin mold.

**Figure 9 polymers-15-01071-f009:**
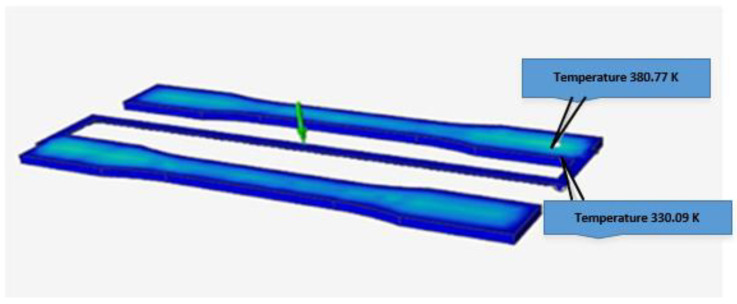
Simulations of two single-point temperatures in a specimen instrumented with a thermocouple: specimen surface (top) and area in contact with the mold cavity and the inserted thermocouple (bottom).

**Figure 10 polymers-15-01071-f010:**
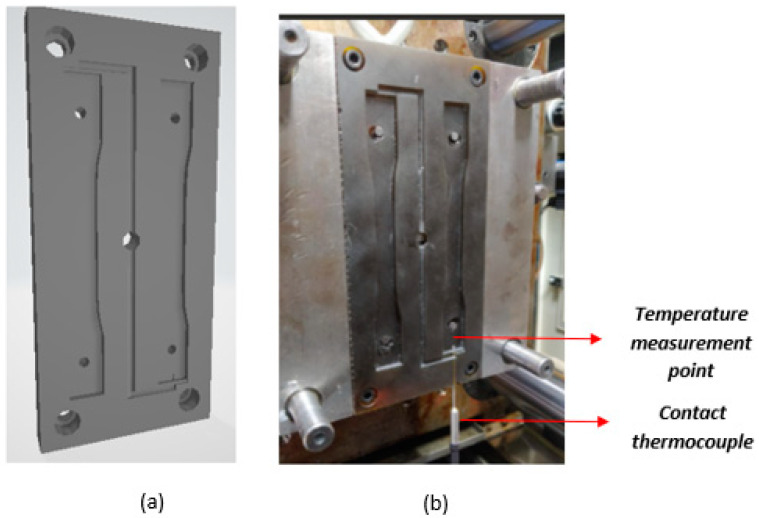
(**a**) CAD of rapid tooling mold for ASTM D638-compliant specimens. (**b**) Mold instrumented with a thermocouple to measure internal temperatures. (**c**) Thermocouple location and specimen measurements (in mm).

**Figure 11 polymers-15-01071-f011:**
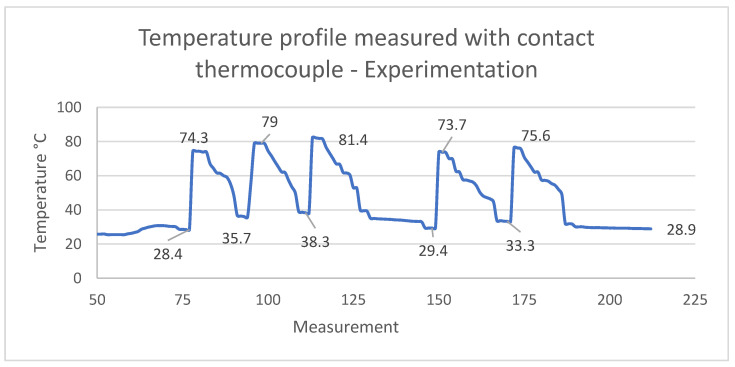
Temperature profile measured by a thermocouple during the experimental injection process.

**Table 1 polymers-15-01071-t001:** 3D printing parameters for the insert mold manufactured by SLA.

3D Printing Parameters	
Printing time (h:m:s)	16:05:51
Layer thickness (µm)	350
Infill (%)	100
Number of layers	1909
Percentage of supports (%)	19
Quantity of resin used	524.88 mL/524.875 g

**Table 2 polymers-15-01071-t002:** Mechanical properties of the HighTemp DL400 resin [18].

**Tensile Properties**		
Tensile Modulus *	3800–4000 MPa	ASTM D638
Ultimate Tensile Strength *	80 MPa	ASTM D638
Elongation at break *	4%	ASTM D638
**Flexural Properties**		
Flexural Strength *	109 MPa	ASTM D790
Flexural Modulus *	3300 MPa	ASTM D790
**Impact Properties**		
Impact Strength Notched Izod *	3.1 KJ/m^2^	ISO 180
Impact Strength Notched Izod *	15.6 J/m	ASTM D256
**General Properties**		
Hardness *	95 Shore D	ASTM D2240
Heat Deflection Temperature *	230 °C	ASTM D648 (0.455 MPa)
Viscosity	650 cPs	At 25 °C Brookfield spindle 3
Density	1.10 g/cm^3^	
Storage	10 < T < 50 °C	
Post-cured for 1 h at 60 °C in Cure L2		

* According to Standard ASTM/ISO.

**Table 3 polymers-15-01071-t003:** Material properties of polypropylene (i.e., a copolymer) [19].

Material Property	Unit	Homopolymer	Copolymer
Density	Kg/m^3^	905	905
Price/Ton	/₤	680	620
Tensile Strength	/Mpa	33	25
Tensile Modulus	/Gpa	1.4	1
Elongation at break	/%	150	300
Hardness	/Rockwell R Scale	90	80
Notched Izod Impact	/KJ/m	0.07	0.1
Heat Distortion Temp (HDT)	@ 0.45 Mpa/°C	105	100
Heat Distortion Temp (HDT)	@ 1.8 Mpa/°C	65	60
Oxygen Index	/%	17	17
Melt Flow Index (MFI)	g/10 min	5.9 ± 0.1	

**Table 4 polymers-15-01071-t004:** Injection parameters used here to mold the specimens according to ASTM D638 standard.

**Injection Parameters**	**Zone 1**	**Zone 2**	**Zone 3**
Position (Distance in mm)	28	15	4.5
Pressure (MPa)	6	5	3
Velocity (%)	60	50	10
**Holding Parameters**	**Zone 1**	**Zone 2**	**Zone 3**
Velocity (%)	30	30	0
Pressure (MPa)	-	1	1
Time (s)	-	4	1

**Table 5 polymers-15-01071-t005:** Technical specifications of the EXTECH SD200 3-Channel Temperature Datalogger (taken from its datasheet) [17].

Specifications	Range	Resolution	Accuracy (% + Digits)
Temperature	−58 to 2372 °C	0.1 °C	+/− (0.5% + 1 °C)
	−100 to 1300 °C		+/− (0.5% + 0.5 °C)
Memory	2000 k data using 2G SD memory card		
Dimensions	132 mm (long), 80 mm (wide) and 32 mm (height)		
Weight	9.9 Oz (282 g)		

**Table 6 polymers-15-01071-t006:** Summary of results of the tensile tests on specimens produced by SLA 3D printing.

Test	Maximum Strength (kN)	Maximum Tensile Stress (MPa)	Breaking Strain (%)	Fracture in Calibrated Zone
T1	1.4	27.2	1.21	Yes
T2	1.58	29.5	1.21	Yes
T3	-	-	-	No
T4	1.57	29.5	1.16	No
T5	1.33	25.1	0.96	Yes
T6	1.14	21.5	0.82	No
Average	1.40	26.6	1.070	-
Standard deviation	0.18	3.4	0.18	-

**Table 7 polymers-15-01071-t007:** Consolidated results of the tensile test on polypropylene specimens obtained in a 3D printed resin mold insert.

Test	Max. Strain (N/mm^2^)
Test 1	27.78
Test 2	28.61
Test 3	28.62
Test 4	27.15
Test 5	28.28
Average	28.09

**Table 8 polymers-15-01071-t008:** Consolidated results of the tensile tests on specimens obtained in a duralumin mold.

Test	Max. Strain (N/mm^2^)
Test 1	26.19
Test 2	24.05
Test 3	23.87
Test 4	24.23
Test 5	23.69
Average	24.40

**Table 9 polymers-15-01071-t009:** Average simulated temperatures in two areas in the cavity.

Average Simulated Temperatures		
Top area	380.77 K	107.62 °C
Bottom area	330.09 K	56.94 °C
Average temperature	355.43 K	82.28 °C

**Table 10 polymers-15-01071-t010:** Experimental temperature measurements in the area of contact between the cavity and the thermocouple.

Measurement	Temperature (°C)
Measurement 78	74.9
Measurement 98	79
Measurement 116	81.4
Measurement 151	73.7
Measurement 174	75.6
Average	76.92

## Data Availability

Not applicable.

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
