# Peer review of "Development, Simulation of Temperatures, and Experimentation in Injection Molds Obtained through Additive Manufacturing with Photocurable Polymeric Resins"

_polymers, 2023, doi:10.3390/polym15051071_

Round 1
Reviewer 1 Report
12/15/2022
Polymers 2022, 14, x. https://doi.org/10.3390/xxxxx
Development, simulation of temperatures, and experimentation in injection molds obtained through additive manufacturing with photocurable polymeric resins
Reviewer’s comments:
The article examines the rapid tooling for the injection process using current additive manufacturing technology via experiments and simulations. Even though the present results show a certain value as a scientific/engineering contribution, I have serious doubts about the presentation of the study.
The reviewer has a few major concerns about the method and results. The authors should also respond to the following comments for further review:
1. A few keywords emphasizing “age of artifacts” can be added.
2. Abstract should be reworded. Please get rid of long sentences. Please clearly explain why, what, how you did and your major results, and how your results changed the world.
3. Introduction required a deeper literature survey.
4. The authors are strongly encouraged to edit the manuscript and remove gibberish throughout the manuscript.
5. I would like to see the details of your experimental method and results.
6. Overall figure qualities can be improved.
7. Can you please explain how practical and easy it is to use your framework in real-time analyses of in-service structures?
8. Can you please explain clearly what the significance of your study is?
Author Response
Response document for the reviewers of the article entitled Development, simulation of temperatures, and experimentation in injection molds obtained through additive manufacturing with photocurable polymeric resins
Manuscript ID: polymers-2114411
General aspects: The authors would like to thank the reviewers for their time, work, and expert comments devoted to correcting our manuscript so that it can have the required technical and scientific quality. Each one of your comments was carefully analyzed and acted upon by revising or correcting the passages. All the changes made to the manuscript are shown below in green.
Reviewer 1
The article examines the rapid tooling for the injection process using current additive manufacturing technology via experiments and simulations. Even though the present results show a certain value as a scientific/engineering contribution, I have serious doubts about the presentation of the study.
The reviewer has a few major concerns about the method and results. The authors should also respond to the following comments for further review:
- A few keywords emphasizing “age of artifacts” can be added.
R./ Following this comment, the keyword “age of artifacts” was added.
- Abstract should be reworded. Please get rid of long sentences. Please clearly explain why, what, how you did and your major results, and how your results changed the world.
R./ The abstract has been reworded. Now, it clearly explains every topic (why, what, how?) according your comments.
Abstract: Additive manufacturing (AM) is a relatively new option in mold manufacturing for rapid tooling (RT) in injection processes. This paper presents the results of experiments with mold inserts and specimens obtained by stereolithography (SLA), which is a kind of AM. A mold insert obtained by AM and a mold produced by traditional subtractive manufacturing were compared to evaluate the performance of the injected parts. In particular, mechanical tests (in accordance with ASTM D638) and temperature distribution performance tests were carried out. The tensile test results of specimens obtained in a 3D printed mold insert were better (almost 15%) than those produced in the duralumin mold. The simulated temperature distribution closely matched its experimental counterpart—the difference in average temperatures was merely 5.36 °C. These findings support the use of AM in injection molding and RT as an excellent alternative for small and medium-sized production runs in the global injection industry.
- Introduction required a deeper literature survey.
R./ A deeper literature review was incorporated into the Introduction (Section 1), and the state of the art is reviewed in Section 2, including more references about this topic.
Introduction (lines 53–71)
Another study investigated the production of injection molds (prototypes) by three different technologies: stereolithography (SLA), laser sintering (LS), and 3D resin photopolymerization (e.g., PolyJet). To validate the use of these three technologies in injection molding, different materials were evaluated to find the most suitable option for each [2].
Other authors implemented SLA with photocurable resins to achieve fast injection tooling and evaluate the mechanical and thermal properties of a commercial polypropylene. As their results indicate that the nature of the resin inserts affects the crystallinity of the mold parts in terms of microstructure, their recommendation is to use the molds in short production runs and pilot tests [3]. The influence of thermomechanical loads on the life cycle of mold inserts produced by additive manufacturing can be a differential factor that should be studied to improve the performance of these tooling technologies [4].
Different studies have characterized material microstructures in order to create simulation models based on information about material properties obtained by multiple techniques, e.g., differential scanning calorimetry (DSC), thermomechanical analysis (TMA), and scanning electron microscopy (SEM). To elaborate on the ideas presented in this introduction and delve deeper into the literature on this topic, the next subsection describes the state of the art of SLA in the industry of RT mold injection.
And section 2 – The state of the art of the SLA technique in rapid tooling (lines 78–92)
Some of them evaluated the performance of molds made with reticulated structures based on titanium, aluminum, and vanadium alloys. In their case, they used computer-aided engineering (CAE) simulations to observe the behavior of these light-weight structures (up to 80% lighter than traditional molds). In their experiments, they achieved 400 complete production cycles in an injected polyvinyl chloride (PVC) material [11].
Others investigated the microtexture of molds obtained by SLA that are employed to manufacture medical components, evaluating the effect of SLA on mold surfaces. They studied the tribological properties of said microtexture and their impact on some biomedical implants and found that the printing direction can improve the surface properties of the piece [12].
Zirconia has also been used as a stabilizing material for molds obtained by SLA [6] and its derivative, i.e., digital light processing (DLP), which is the SLA approach adopted in this study. Nowadays, several industrial processes are already using additive manufacturing to obtain molds for short production runs and cycles as an alternative to tooling by traditional manufacturing [3], [13], [14].
- The authors are strongly encouraged to edit the manuscript and remove gibberish throughout the manuscript.
R./ The entire manuscript was edited by a language professional. The grammar was corrected, and its quality was improved in several regards.
- I would like to see the details of your experimental method and results.
R./ In the link below, you can find details, videos, files, and tables supporting our experimental methods and results.
Methodology and experimentations - OneDrive (sharepoint.com)
- Overall figure qualities can be improved.
R./ The quality of Figures 1, 2, 3, and 9 was improved. In particular, Figures 1 and 9 were replaced.
- Can you please explain how practical and easy it is to use your framework in real-time analyses of in-service structures?
R./ Interchangeable mold inserts were manufactured by stereolithography. This method is practical because you can design simple or complex geometries for cavities in stereolithographic resin inserts and put them in a structure or framework for injection molds and easily assemble them in the machine. This process can reduce assembly times and the production time of different formats and sizes in batch production. Real-world experiments with interchangeable inserts in an injection machine yielded positive results, which could mark the beginning of new production systems in the injection industry.
- Can you please explain clearly what the significance of your study is?
R./ The following highlights stress the significance of our study for the injection molding industry:
- Additive manufacturing (SLA in particular) can be used to design easy, medium, and complex cavities for the injection molding industry.
- SLA achieved good results for rapid tooling (mold inserts) in shorter production and manufacturing times than traditional and subtractive manufacturing.
- Designing and manufacturing interchangeable inserts can reduce the time it takes to change formats according to the SMED (Single-Minute Exchange of Die) philosophy.
- The results indicate that the mechanical properties of the part were improved, which is very important for its final quality and product lifecycle.
The following sources were added to the References section:
[2] M. E. Society, “Innovative advances in additive manufactured plastic injection series plastic injection series,” Procedia Manuf, vol. 13, pp. 732–737, 2017, [Online]. Available: https://doi.org/10.1016/j.promfg.2017.09.124
[3] M. H. Jnr et al., “Stereolithography (SLA) utilised to print injection mould tooling in order to evaluate thermal and mechanical properties of commercial polypropylene,” Procedia Manuf, vol. 55, no. C, pp. 205–212, 2021, doi: 10.1016/j.promfg.2021.10.029.
[4] L. Bogaerts et al., “Influence of thermo-mechanical loads on the lifetime of plastic inserts for injection moulds produced via additive manufacturing,” Procedia CIRP, vol. 96, pp. 109–114, 2020, doi: 10.1016/j.procir.2021.01.061.
[11] S. J. Park et al., “Lightweight injection mold using additively manufactured Ti-6Al-4V lattice structures,” J Manuf Process, vol. 79, no. December 2021, pp. 759–766, 2022, doi: 10.1016/j.jmapro.2022.05.022.
[12] V. Basile, F. Modica, R. Surace, and I. Fassi, “Micro-texturing of molds via Stereolithography for the fabrication of medical components,” Procedia CIRP, vol. 110, no. C, pp. 93–98, 2022, doi: 10.1016/j.procir.2022.06.019.
[13] M. A. León-Cabezas, A. Martínez-García, and F. J. Varela-Gandía, “Innovative advances in additive manufactured moulds for short plastic injection series,” Procedia Manuf, vol. 13, pp. 732–737, 2017, doi: 10.1016/j.promfg.2017.09.124.
[14] C. Whlean and C. Sheahan, “Using additive manufacturing to produce injection moulds suitable for short series production,” Procedia Manuf, vol. 38, pp. 60–68, 2019, doi: 10.1016/j.promfg.2020.01.008.

Reviewer 2 Report
This article mentions about the application of 3D printing in manufacturing the mold insert. The SLA technical was applied for the 3D printing process. In general, the content is enough for publishing. However, these issues have to be checked:
1. All figure quality must be improved (Example: Fig. 1, 2, 3, 4, 5...)
2. Part 2 (The stage of the art of the SLA...”) should be integrated into part 1 (Introduction)
3. At the end of the introduction of SLA, paper should have a short conclusion about the SLA technical, as well as the new idea or new application in injection mold, which will use in this research
4. Line 93: “Layer thickness .... 350 µm): in this research, the layer thickness is 350 µm? Is there any reason for selecting this value?
5. The parameters of 3D printing should be select and explain this selection of these values.
6. Table 3: Authors have to explain why select these injection mold parameters for researching, and why these parameter range are chosen.
7. Figure 3 is not clear. Author should show out the layout of the mold cavity and describe about this design
8. Table 4 should have a reference. The unit of temperature should be oC
9. This article has the simulation content, so, author have to show out the simulation model, meshing model and the simulation boundary conditions, as well as the information of simulation software
10. Please explain why using the tensile test for testing the DL400 strength. If this material is used for parts of injection mold, these parts will be pressed or strained in the molding cycle?
11. Line 162: what is “Test tube number 3”? What is “tube”?
12. Table 5:
- How many specimens for 1 testing case?
- What is the difference of printed parameters between these cases?
13. Add some discussion and explanation of Table 5
14. Figure 5b: The molding part has the flash trouble; authors should change other case in this Figure
15. Line 198 – 204: The crack in Figure 6 appeared after 85 cycles, so, how the better the mold insert is?
16. Table 6 and 7: What are the manufacturing conditions of these samples (Test 1,2,3,4...)?
17. Please discuss and explain the results of part 4.3
18. Line 24: where are the locations of thermocouple?
19. Figure 9 should be clearer
20. Line 251 – 255: When is these temperatures cached? (End of filling, end of packing, end of cooling...)
21. This article should have a figure to show out the dimension of cavity plate, as well as the position of sensors. (Figure 10 is not clear for observing these positions)
22. Line 265 – 269: data loading time is 5s, this time step id too large, do you have any reference or any reason for using this time step?
23. Figure 11: Check the word “Medicion”
24. Add more references
25. In general, this paper has to be modified the format
Author Response
Response document for the reviewers of the article entitled Development, simulation of temperatures, and experimentation in injection molds obtained through additive manufacturing with photocurable polymeric resins
Manuscript ID: polymers-2114411
General aspects: The authors would like to thank the reviewers for their time, work, and expert comments devoted to correcting our manuscript so that it can have the required technical and scientific quality. Each one of your comments was carefully analyzed and acted upon by revising or correcting passages. All the changes made to the manuscript are shown below in red.
Reviewer 2
This article mentions about the application of 3D printing in manufacturing the mold insert. The SLA technical was applied for the 3D printing process. In general, the content is enough for publishing. However, these issues have to be checked:
- All figure quality must be improved (Example: Fig. 1, 2, 3, 4, 5...)
R./ The quality of Figures 1, 2, 3, and 9 was improved. In particular, Figures 1 and 9 were replaced.
- Part 2 (The stage of the art of the SLA...”) should be integrated into part 1 (Introduction)
R./ A deeper literature review was incorporated into the Introduction (Section 1), and the state of the art is reviewed in section 2 of the paper, including more references about this topic.
- At the end of the introduction of SLA, paper should have a short conclusion about the SLA technical, as well as the new idea or new application in injection mold, which will use in this research
R./ According to this suggestion, the following passage has been included at the end of Section 2 (lines 114–122):
As described in Sections 1 and 2, SLA is very important for the injection industry because it can produce easy, medium, and complex geometrical designs and cavities for injection molding. It can also be used to develop and manufacture interchangeable inserts, which can reduce the time it takes to change formats according to the SMED (Single-Minute Exchange of Die) philosophy. Based on the results reported in Section 4 of this paper, the mechanical properties of the parts were improved, which is very important for final quality and product lifecycle. All these aspects had not been considered together in any of the articles reviewed in this section, which constitutes the research gap addressed in this study.
- Line 93: “Layer thickness .... 350 µm): in this research, the layer thickness is 350 µm? Is there any reason for selecting this value?
R./ The Materials and methods section now explains why a 350-µm layer thickness was selected. More specifically, the reason is that the thicker the layer, the shorter the manufacturing time (while still producing good results). The following explanation was added.
Lines 139–145
Different layer thicknesses (50, 100, 200, and 300 µm) were tested, and they produced the same result in surface quality and good definition. However, low layer thicknesses require longer manufacturing times. Therefore, a layer thickness of 350 µm was selected to achieve printed parts in shorter times, which makes this process a suitable alternative for fast tooling and developing injection mold inserts. The 3D printing parameters in Table 1 were selected to achieve the best quality in the mold insert (i.e., good tolerance in the surface of the piece, good filling) in the shortest possible time.
- The parameters of 3D printing should be select and explain this selection of these values.
R./ A new table (Table 1) was added to specify the 3D printing parameters. In addition, the selection of the specific parameter values is now explained in lines 143 to 145.
- Table 3: Authors have to explain why select these injection mold parameters for researching, and why these parameter range are chosen.
R./ It was added an explanation for choosing the injection mold parameters, as well as an important parameter or characteristic for the molten material (polypropylene), the melt flow index (MFI), which is included in the last row of table 3, which determines every parameter in the screw injection machine. Please review it between lines 167 and 176.
By adjusting these parameters according to the results of preliminary injection tests, it is possible to obtain a complete filling of the mold cavity with homogeneous temperatures of the molten plastic. The holding parameters make it possible to inject a piece without noticeable contractions and to achieve a constant weight. In this study, these parameters were adjusted using computer simulations and preliminary experimental evaluations of injected parts’ weight.
- Figure 3 is not clear. Author should show out the layout of the mold cavity and describe about this design
R./ Figure 3 was improved and should be clearer now. The layout of the mold cavity is now described in lines 187 to 191:
The cavity mold was designed to produce two parts for tensile testing in a single injection cycle. Figure 5a shows the runner and feeding system. The feeding system parts do not interfere with the quality of the piece because they are located at the ends of the specimen (top and bottom on the tensile tester), that is, the areas of contact between the specimen grips of the tensile testing machine and the part.
- Table 4 should have a reference. The unit of temperature should be oC
R./ The temperature unit was changed to °C, and the respective reference was included.
- This article has the simulation content, so, author have to show out the simulation model, meshing model and the simulation boundary conditions, as well as the information of simulation software.
R./ Following this comment, the simulation software (Altair® Inspire™ Mold) is now specified in line 210, and more comprehensive information about the simulation model is now included in lines 213 to 217.
- Please explain why using the tensile test for testing the DL400 strength. If this material is used for parts of injection mold, these parts will be pressed or strained in the molding cycle?
R./ We conducted the tensile tests only to provide a point of reference for the mechanical properties of the DL400 resin. The data sheet of this resin specifies a maximum tensile strength of 80 MPa before processing (more information: https://photocentricgroup.us/es/product/hightemp-dl400/). According to the results in Table 6, its maximum tensile strength after 3D printing was 26.6 MPa. Therefore, it was very important to compare the tensile strength of the three kinds of specimens obtained in this study: (i) DL400 resin specimens obtained by SLA, i.e., additive manufacturing; (ii) polypropylene specimens obtained in 3D printed DL400 polymer resin mold inserts; and (iii) polypropylene specimens obtained by injection molding inside a traditional duralumin mold produced by subtractive manufacturing.
- Line 162: what is “Test tube number 3”? What is “tube”?
R./ It should have been “specimen.” This mistake was corrected throughout the text. For example, lines 232–233 now read:
“Test specimen 3 (T3), in Table 6, presented a very low tensile stress value (an outlier), probably due to the brittle behavior of the resin in the grip area.”
- Table 5:
- How many specimens for 1 testing case?
R./ ASTM D638 requires five specimens, but we tested six in total. Their results are now presented in Table 6 (which was “Table 5” in the previous version of the manuscript).
- What is the difference of printed parameters between these cases?
- Add some discussion and explanation of Table 5
R./ Table 6 (“Table 5” in the previous version) is now explained in a paragraph below it (lines 240–248).
- Figure 5b: The molding part has the flash trouble; authors should change other case in this Figure
R./ Maybe the intention of including that figure was not clearly stated before. We wanted to show some problems that arise in injection molding experiments and that should be solved with continuous testing. Figure 5b represents one of those problems (i.e., flash) that occur in the short term. Future studies and experiments in the field can address said problem. This is now explained in lines 252 to 253 (about Figure 5b).
- Line 198 – 204: The crack in Figure 6 appeared after 85 cycles, so, how the better the mold insert is?
R./ To the best of our knowledge, this is the first report of experimental tests with mold inserts produced by SLA. This number (85 cycles) represents the best performance that we could achieve with this insert, without any successful results before this one. This is an original study that will be furthered with future and continuous experimentation.
- Table 6 and 7: What are the manufacturing conditions of these samples (Test 1,2,3,4...)?
R./ The manufacturing conditions for the specimens produced by 3D printing and those injected into a duralumin mold are now presented in Tables 1 and 4 (new table), respectively. Please review them again.
- Please discuss and explain the results of part 4.3
R./ More discussion and conclusions were added to Subsection 4.3 (lines 321–331).
- Line 24: where are the locations of thermocouple?.
R./ The thermocouple was located at the bottom of the right cavity (line 201).
- Figure 9 should be clearer.
R./ Figure 9 was improved and should be clearer now.
- Line 251 – 255: When is these temperatures cached? (End of filling, end of packing, end of cooling...)
R./ The simulation and experimentation conditions described in Section 3 represent the cooling stage of the mold (see lines 350–351).
- This article should have a figure to show out the dimension of cavity plate, as well as the position of sensors. (Figure 10 is not clear for observing these positions)
R./ Thanks for this recommendation. A new figure (Figure 10c) was added to show the overall dimensions of the cavity plate and the position of the thermocouple.
- Line 265 – 269: data loading time is 5s, this time step id too large, do you have any reference or any reason for using this time step?
R./ This time step was used because the datalogger employed in the experimentation has only one setting for the measurements, that is, every 5 seconds.
- Figure 11: Check the word “Medicion”
R./ Thanks for the correction. “Medición” was replaced with “Measurement” in Figure 11.
- Add more references
R./ Eight more references were added, for a total of 23.
- In general, this paper has to be modified the format
R./ Thank you for your comments and corrections. Following your recommendation, the entire manuscript was edited by a language professional in terms of grammar and writing quality. As a result, the manuscript was improved in several regards, including the format.

Round 2
Reviewer 1 Report
The authors have addressed my comments and questions. The paper is recommended for publication in the Polymer journal.
Important note: Please disregard my first comment in my first review. This was unintentionally included in the review. My apologies for the confusion.
Author Response
Second response document for the reviewers of the article entitled ´Injection molds obtained by additive manufacturing using photocurable polymeric resins: Development, temperature simulation, and experimentation´.
Manuscript ID: polymers-2114411
General aspects: The authors would like to thank the reviewers for their time, work, and expert comments devoted to correcting our manuscript so that it can have the required technical and scientific quality. Each one of your comments was carefully analyzed and acted upon by revising or correcting the passages. All the changes made to the manuscript are shown below in green.
Reviewer 1
The authors have addressed my comments and questions. The paper is recommended for publication in the Polymer journal.
Important note: Please disregard my first comment in my first review. This was unintentionally included in the review. My apologies for the confusion.
R./ Thank you for your comments. It was necessary to improve our work. Regarding the first comment, this keyword was deleted.

Reviewer 2 Report
Dear authors,
In general, this version was improved clearly. However, I think there are some points that you should check. So, please reply this issue:
1. Table 5 (Table 6 in the last version): - What is the difference of printed parameters between these cases?
2. Do you have any reason or any explanation about the location of thermal couple? (Figure 10) Please add more information about the thermal couple, as well as the assembly method of this sensor into the mold base.
3. Figure 10: the number is not clear
4. Table 10: what is the meaning of measurement 78, 98, ...?
Sincerely yours,
Author Response
Second response document for the reviewers of the article entitled ´Injection molds obtained by additive manufacturing using photocurable polymeric resins: Development, temperature simulation, and experimentation´.
Manuscript ID: polymers-2114411
General aspects: The authors would like to thank the reviewers for their time, work, and expert comments devoted to correcting our manuscript so that it can have the required technical and scientific quality. Each one of your comments was carefully analyzed and acted upon by revising or correcting passages. All the changes made to the manuscript are shown below in red.
Reviewer 2
Dear authors,
In general, this version was improved clearly. However, I think there are some points that you should check. So, please reply this issue:
- Table 5 (Table 6 in the last version): - What is the difference of printed parameters between these cases?
R./ There are no differences between the 6 cases; the printed parameters are the same because 6 tensile plates were printed in the same production batch. There must be at least six samples for the ASTM Standard 638 test. This was done according to the standard specifications mentioned.
- Do you have any reason or any explanation about the location of thermal couple? (Figure 10) Please add more information about the thermal couple, as well as the assembly method of this sensor into the mold base.
R./ For ease of location and proximity to the injection point. In addition to this, the thermal couple was placed at the entrance of the cavity, which made it possible to monitor the progression of the temperature in that place throughout the injection cycle of the plastic part. In this way, the thermal couple record provides a graph of various temperature peaks over time (Figure 11). The difference between each temperature peak represents one injection cycle, which includes the time required to inject, cool, and eject the created plastic part.
- Figure 10: the number is not clear
R./ Figure 10's numbers have been improved and are now clear. Please review this.
- Table 10: what is the meaning of measurement 78, 98, ...?
R./ They are the sequences in which there were significant measurements at high temperatures. Each measurement was made at a frequency of 5 seconds, as shown in Figure 11. These measurements (78.98...) correspond to the physical temperature in each injection cycle achieved when the thermocouple located in the area captured the highest temperature, that is, when the molten material entered the cavity.

Round 3
Reviewer 2 Report
Dear Authors,
In general, I think this version is good for publishing. However, if it is possible, you should improve the quality of these Figure: 3, 9, and 10c
In addition, the paper format is needed to be refine.
Sincerely yours,